# Transtheoretical Model for Physical Activity in Older Adults: Systematic Review

**DOI:** 10.3390/ijerph17249262

**Published:** 2020-12-11

**Authors:** Fabio Jiménez-Zazo, Cristina Romero-Blanco, Nuria Castro-Lemus, Alberto Dorado-Suárez, Susana Aznar

**Affiliations:** 1PAFS Research Group, Faculty of Sports Sciences, University of Castilla-La Mancha, 45071 Toledo, Spain; fabio.jimenez@uclm.es (F.J.-Z.); alberto.dorado@uclm.es (A.D.-S.); 2PAFS Research Group, Faculty of Nursing, University of Castilla-La Mancha, 13071 Ciudad Real, Spain; cristina.romero@uclm.es; 3FENIX Research Group, Faculty of Sports Sciences, University of Sevilla, 41004 Sevilla, Spain; ncastro@us.es

**Keywords:** transtheoretical model, older adults, physical activity

## Abstract

Healthy aging makes the practice of physical activity (PA) a necessity. However, PA guidelines achievement in older adults is scarce. The use of behavioral theories such as Transtheoretical Model (TTM), helps in older adults PA promotion. The aim of this review was to identify the use of TTM for PA in older adults (>60 years). PubMed, SPORTdiscus, and Medline databases were used to conduct the search. All steps of the process followed the recommendations of the PRISMA flow-diagram. We identified eight studies: Six were descriptive cross-sectional studies, one prospective-cohort study and one with a quasi-experimental design. Only two papers evaluated the four behavior change dimensions within the same study, three evaluated the processes of change and the decisional balance, four evaluated the exercise self-efficacy and all assessed the stages of change for PA behavior. From this review, we can conclude that TTM is a useful and suitable behavior model in creating, developing, and evaluating interventions with the aim of acquiring and improving PA habits in the older adults. However, there is paucity of research in this area, and more studies including the four behavioral change dimensions are needed to analyze the effect of TTM on the promotion of PA in the older adults.

## 1. Introduction

Physical activity (PA) is considered as one of the most effective strategies to promote healthy aging [1]. Regular PA practice in older adults is associated with improvements in functional fitness (maintains muscle strength and balance), chronic diseases risk prevention (coronary heart disease, diabetes, and stroke), improvement of mental capacity (self-esteem, maintenance of the cognitive function, reduce anxiety, and depression), and improvement in social response [2,3,4,5,6,7]. PA is a good predictor of healthy aging and decreases the probability of disability in people over 65 years [8].

Recommendations for adults and older adults state the need to achieve at least 150 min of PA at moderate intensity per week, together with muscle strength activities for an improvement in bone mass and activities to improve flexibility, at least 2–3 days per week [9]. 

However, despite the important health benefits of PA, a little proportion of the population over 65 years meet World Health Organization (WHO) PA guidelines [10]. In fact, this group is the least physically active out of all [11]. According to WHO [12], having a sedentary lifestyle is the fourth primary risk factor of non-communicable diseases. The good news are that those who practice regular PA can reduce their risk of pathologies at a rate between 20 and 30% [1].

Older adults’ exercise adherence is a multifactorial process, influenced by: Program characteristics (preference of exercise type), personal factors (demographic, health related, physical, and psychological factors) [13,14], social determinants, and environmental factors [15]. It is important that interventions to promote physical exercise in older adults are based on theories that explain the behavior change, tailored to individual characteristics of the participants [15]. Theory-based interventions to promote PA behavior seem to have a more successful effect than interventions without an established theoretical base [16,17]. In addition, interventions based on a single theory reported greater impact on PA behavior than those interventions with a combination of theories [17].

One of the models that better understands and predicts the behavioral–cognitive changes in the adoption of PA/exercise behavior, is the Transtheoretical Model of Change (TTM), proposed by Prochaska and Di Clemente [18]. The TTM is a most comprehensive and integrated model of behavioral changes among health behavioral models [19], this model began to be used in smoking cessation studies [18], but over time its use has extended to the study of healthy lifestyle promotion, including PA, to become one of the most popular models to understand the PA behavior [20]. This model is a cyclical model and explains the behavior change as a dynamic process, through a temporal dimension by describing them as a sequence of stages and processes by which the individual progress to adapt a regular behavior [21].

The model is composed of different dimensions, summary (Table 1):

Stages of change (SoC): The SoC are the levels of motivation to change, from no intention to change to actually making behavior change. This construction does not explain when people change, rather where people are in their motivation to change. There exist five stages of readiness for change: Pre-contemplation, contemplation, preparation, action, and maintenance [18]. Different studies have shown the relationship between PA levels and SoC, and advanced SoC (action and maintenance stages) tends to present better levels of PA [23,24,25,26].

Processes of change (PC) describe how people change. The PC are the strategies and techniques that people use to change or modify their behavior. The PC can be divided in two categories: Cognitive processes (thinking, attitudes, awareness) and behavioral processes (actions). All processes are important for all SoC, the pattern of use of cognitive versus behavioral PC across the SoC does not change by the stages in PA behavior, in other words, it seems that all PC can be used indistinctively regardless the particular SoC [25,26]. Both behavioral and cognitive PC act together, increasing their combined use across SoC [25,27]. This construct has been considered as one of the key moderators of TTM-based interventions to improve PA behavior [28]

Decisional Balance (DB) determines the belief that people have about the advantages and disadvantages (pros and cons) of making a behavior change, this construct is derived from decisional-making theory [29]. The DB compare the positive and negative aspects of change, differences in the DB, tends to corresponds to different SoC, people in earlier SoC (precontemplation and contemplation stages) perceive more barriers than benefits and people in the later stages (preparation, action, and maintenance stages) perceive more benefits than barriers for change [30].

Self-Efficacy (SE), is the ability to perform specific behaviors in specific situations. This dimension was adapted from Bandura’s Social-Cognitive Theory in 1977 [31]. Exercise SE is an important construct: It is a predictor and a moderator of the adoption and maintenance of PA behavior [28,32]. People with high score in exercise SE tend to be more active [22]. High levels of exercise SE leads to sustain and increase levels of PA in older adults [33]. 

There is a need to understand the adoption and maintenance of exercise behavior in older adults, since the current levels of PA participation of this group are insufficient compared to the positive effects derived from their practice [10]. The TTM is a useful tool within this procedure, since it provides us a detailed vision of the process of acquisition of a new habit. This model has demonstrated to be one of the most effective models within the process of behavior change, with a significant impact in the promotion of PA [17]. Up to date, the TTM has been applied for different behaviors and in different age groups, but there is paucity of research in promoting PA in older adults. Therefore, the objective of this review is to identify the relationship between the different TTM constructs: Stages of change, processes of change, decisional balance, and self-efficacy in relation to PA in the population over 60 years. This review will increase the knowledge about the recent applications of TTM in the promotion of PA in the group of people over 60 years.

## 2. Materials and Methods 

The purpose of this review was to assess the breadth of scientific evidence linking the behavior change and cognitive change referred to adherence to PA practice in adults over 60 year. To accomplish this; we reviewed the impact of use of TTM and their dimensions (stages of change, processes of change, decisional balance, and self-efficiency) and the levels of PA in this age group. 

### 2.1. Information Sources and Search Strategies

We used PubMed, SPORTdiscus, and Medline as electronics databases to get the literature search, we applied search strategies using the following search key words: For the population (elderly), context (Transtheoretical model), and outcome (physical activity). We used a filter to search all fields: ((Transtheoretical Model) AND Elderly) AND Physical Activity.

The literature search was carried out from July to December 2019. From all articles resulting from the search, the abstracts were carefully read, and relevant studies were selected and reviewed in full. To determinate a relevant study, at least three of the researches (FJZ, ADS, CR) had to confirm the validity of the theme of the article independently, in case of divergence, a fourth researcher (SAL) was invited to decide whether to include or exclude the studies. All authors participated in the analysis and interpretation of findings for each topic. A bias risk calculation was included in the process. All authors work individually first prior to elaborate the main results/findings. Furthermore, an adaptation of the STROBE checklist was used [34]. We used it to evaluate the quality of the articles included in our study. STROBE checklist contains 15 items of analysis. The cut-off points established for quality classification were as follows: 0–5 points = low quality; 6–10 points = moderate quality; and 11–15 points = high quality. Studies that achieved high quality, according to the quality criteria, were included in the final analysis.

### 2.2. Study Eligibility Criteria

We only included studies written in English, TTM-based interventions only, older adults over 60 years, and levels of PA. One exception was made to the age criteria due to the WHO [35] considers the ages between 60 and 65 to be the beginning of old age in most territories. We did not include studies that made use of another theory based models different to TTM to explain the PA behavior change, as well as studies with subjects under the age of 60 years and those studies which no subject age specified. Finally, we excluded from consideration, studies of other topics, duplicated studies, protocols, review, or metanalysis, and studies published before 2014. All the steps of the process followed the recommendations of the PRISMA Flow Diagram [36], as presented in Figure 1.

### 2.3. Data Items

To classify commonality between articles, data was collected and organized by year of publication, journal, author, country/territory, intervention design, study population, population characteristics, outcomes of measures, and key findings.

## 3. Results/Overview

Figure 1 details all the steps of the processes followed according to the recommendations of the PRISMA flowchart in the studies selection. We identified a total of eight studies published between 2014 and 2019 which were included in the systematic review from the original 147 papers identified by the literature search [37,38,39,40,41,42,43,44]. The most common reason for excluding studies was that they did not meet the inclusion criteria, mainly by the age. 

The main characteristics, TTM dimensions, outcomes, and key findings for each of the 8 articles included in this review are listed in Table 2 and Table 3. Of these eight articles, six studies were descriptive-cross-sectional studies [37,38,39,40,41,44], one was a prospective cohort study [42] and one was a quasi-experimental design [43]. Two of the studies originated from Ireland [37,41], two more from Taiwan [42,43] and individual studies from Iran [38], Korea [39], Japan [44], and Italy [40]. According to the study population, the review gather a total of 2304 participants, included both men (51.65%) and women (48.35%). Only three of the studies (37.5%) worked with healthy older adults [38,43,44] and the rest with pathologies (62.5%) (bronchiectasis [37], non-cystic fibrosis bronchiectasis [41], physical disability or brain injury [39], type II diabetes [40], and patients after open heart surgery [42]). Regarding to TTM-interventions characteristics, we can find the following topics: Measuring PA levels [40,42], to identified daily patterns of PA [41], to understand the levels of PA and sedentarism [37], to identify knowledge of muscle strength exercise recommendations [44] to create PA-interventions tailored to specific SoC [39,43] and to corroborate the usefulness of this model in older adults [38].

In the following sections, we “drill down” in the different constructs of TTM to examine what has been found regarding the influence TTM on PA.

### 3.1. Stages of Change (SoC)

The SoC have been considered as predictors of PA [40]. Significant differences (*p* < 0.05) were found between the minutes of PA/week [40,41,42], daily total PA time, daily light-lifestyle PA time, daily total moderate to vigorous physical activity (MVPA) time and daily steps counts [41] according to the different SoC. The average of bouts (10-min) MVPA and activity energy expenditure increased and the sedentary behavior time decreased regarding to SoC but not significantly [41]. Moreover, there was an increase in PA levels and a decrease of sedentary behavior time as we moved through to more advanced SoC (action and maintenance stages) [40,41,42]. Finally, advanced stages of behavior change corresponded to greater benefits and lower perceived barriers referred to the achievement of muscle strength exercise recommendations [44]. 

### 3.2. Processes of Change (PC)

The PC are the techniques and strategies that people use to change or modify their behavior. PC were clearly identified in patients with bronchiectasis, showing a likewise use of cognitive and behavioral processes in their PA behavior [37]. In patients with non-cystic fibrosis bronchiectasis, it was observed as initial SoC (precontemplation and contemplation stages) used significantly fewer PC compared to more advanced SoC (action and maintenance stages) (*p* > 0.01), on the other hand, precontemplation and contemplation stages tended to use cognitive PC (*p* = 0.031), preparation stage used behavioral and cognitive PC equally (*p* = 0.92) and action and maintenance stages tended to use of behavioral PC (*p* = 0.055) [41]. In patients with physical disability or brain injury, no significant differences were found among PC of ER, CC, HR, and RM across the SoC (*p* > 0.05), while, the patterns of use PC of CR, DR, SR, SL, SeL, and SC varied significantly across precontemplation, contemplation, and preparation stages (*p* < 0.05) [39]. Finally, the use of PC to create and tailor interventions with the aim of promoting PA in the older adults, favored the progression of participants to more proactive SoC (action and maintenance) and therefore improved the acquisition of healthy habits [43].

### 3.3. Decisional Balance (DB)

There were a significant differences (*p* < 0.001) between the DB score among the different SoC [38]. The DB can be divided in two dimensions: Benefits and barriers. According to benefits, there was a direct association (*p* < 0.001) between perceived benefits and PA levels [38] and an inverse association (*p* = 0.009) with sedentary time [37]. Significant differences were observed between SoC and perceived benefits (*p* < 0.001) [38]. Regarding the barriers, we observed an inverse correlation (*p* < 0.001) between perceived barriers and levels of PA, in addition, there were significant differences between SoC and perceived barriers (*p* < 0.001) [38]. In patients with bronchiectasis [41] there were no significant differences between the DB score (*p* = 0.31) and the benefits perceived (*p* = 0.92) with the SoC.

### 3.4. Self-Efficacy (SE)

The SE refers to people’s confidence in their ability to make a change in specific situations. Two of the articles considered SE as predictor of PA [38,40]. Significant differences were found between the SE score and the SoC (*p* < 0.01) [38,40]. In addition, a direct association between SE and PA levels was observed (*p* < 0.01) [38,40]. Advanced SoC (action and maintenance stages) had higher SE scores [38] with a linear trend (*p* < 0.001) in the result [40]. On the other hand, studies with older adults with respiratory pathologies did not find any significant differences (*p* = 0.14) between SE score and SoC [41] and they reported that their SE was reduced when they experienced situations related to symptoms of their disease [37].

## 4. Discussion

The aim of this review focused on identifying the recent applications of TTM (2014–2019) and the relationship between its different constructs (stages of change, processes of change, decisional balance, and self-efficacy) with the levels of PA in the older adults over 60 years old. After completing all the recommended procedures for systematic reviews (PRISMA flow-diagram, conference by three researchers independently and finally quality of articles checklist adapted from the STROBE instrument), 8 original articles were included in the analysis.

This is the first study that reviews the use of the TTM for physical activity in older adults. The study encourages the use of TTM for physical activity in clinical practice as part of the older adults health screening at General Practice practices. Moreover, the use of TTM for physical activity can also lead to more suitable PA programs and/or intervention aimed to improve healthy aging, precisely because this group is the most vulnerable due to their low levels of PA [45].

Theory-based interventions are more effective to improve PA behavior than non-theories based ones [17]. Similarly, one theory-based approach seem to be more effective than multiple theories approach or a combination of theories [17]. All papers included in this review have included a single theory and have been based on the TTM theory. 

The TTM is composed of differences constructs: SoC, PC, DB and SE [21]. From the eight papers included in our review, only two [37,41] evaluated the entire TTM in older adults. The number of constructs used within an intervention determines its chances of success. A recent meta-analysis determined that using at least three of the constructs from the TTM, improved the chances of increasing PA by three times [28]. The use of more constructs from the TTM has already been reported by previous studies in which it was highlighted that many of the researches are inspired by the theory but they were not theory-driven [26,28]. The evaluation of all the constructs of behavioral model will allow the researchers to understand more precisely the specific characteristics of this population and their behavior change processes, thus favoring the sustainability and adherence of participants to exercise interventions. 

Motivation for change is the basis of this behavioral model (TTM) and it is determined by the SoC. This construct has been evaluated in all the articles of this review, in order to categorize the subjects according to their intention to behavior change and to evaluate the results that occurred throughout an intervention. The relationship between levels of PA and sedentary time with the SoC has already been reported in other reviews [25,26,46], determining that the initial SoC were related to longer periods of sedentary lifestyle and advanced SoC showed higher levels of PA. It is important to highlight how two articles [39,43], focused their research on the initial SoC (precontemplation, contemplation and preparation stages), this approach seems to be an opportunity to focus strategies to promote PA within non-active or sedentary population. Previous researches determined that SoC were not considered moderators of the efficacy of TTM-based interventions in promoting PA behavior, so they suggest not focusing future researches exclusively on SoC [26,27,28]. 

On the other hand, PC are a very useful tool in the design of “tailored” interventions since they will allow us to know the strategies that individuals use to make a behavior change. This construct has been considered one of the successful moderators of TTM-based PA interventions [28]. PC have been measured in 3 of the 8 articles evaluated, encompassing only the population with specific pathologies (bronchiectasis, non-cystic fibrosis bronchiectasis, physical disability and brain injury). In particular, specific recommendations for the precontemplation, contemplation and preparation stages in patients with physical disabilities and brain injury were designed on-demand [39], this work reflects a practical application of this model in the development and adaptation of interventions aimed at the promotion of adapted PA in the older adults. According to the use of PC in relation to SoC, we cannot establish a clear relationship with the results obtained. The current scientific literature suggests that there are no differences between patterns of use of PC and SoC [26], the cognitive and behavioral PC act together, increasing their use across to SoC [25,27]. In addition, It has been observed how the use of PC in PA interventions doubles the chances of success in the acquisition of PA behavior [28].

SE has been the second most evaluated variable after the SoC. Self-efficacy provides us with information about the ability that individuals have to maintain behavior when faced with diverse and difficult situations. Within the review, SE has been measured both in a healthy and with pathologies population, considering 2 of the 4 articles that evaluated it as a predictor of the practice of PA [38,40]. These results supported the existing literature on the need to enhance SE to favor the adherence of older adults to PA promotion programs [26,28,33].

Another construct of TTM in the promotion of PA in the older adults, are all those implicit and explicit decisions that the individual weighs (i.e., DB). In this review, we have seen that it has been evaluated in 3 of the 8 selected articles [37,38,41], where it has been reflected that older adults perceive more benefits and lower barriers in the stages of action and maintenance against the initial stages. Knowing the aspects that the individual perceives as potentially favorable or dangerous in their practice of PA, will help the development and sustainability of interventions focused on the encouragement and promotion of PA in the older adults.

The group of older adults is considered a heterogeneous group because of the great variety of characteristics among them. Therefore, it would be interesting to explore other mediating and moderating variables of the exercise interventions [47,48] such as social support, outcome expectation, perception of the environment, and resilience among others, in order to achieve a more complete vision of this process [49]

Our findings are in line with the results reported by recent reviews focusing on TTM-interventions to promote PA in adults [26,27,28], our results contribute to increase the evidence about the positive effect of this behavior model in promoting healthy habits of PA in older adults, but more research is needed to understand the use of PC, their relationship with SoC and their relationship in the process of promoting and maintaining PA behavior in people over 60 years old. In addition, more studies encompassing the entire constructs of TTM are needed to fully understand PA behavioral change in this population. Finally, it would be interesting to extend the use of TTM to promote PA behavior to other groups of older adults with specific characteristics or pathologies.

The present study has some limitations. Firstly, many of the studies did not evaluate all the constructs of the behavioral model, so that we do not have enough information to understand its true effectiveness. Secondly, the use of the language restriction as an exclusion criterion (only documents in English) could limit the articles found. Finally, it is possible that relevant TTM-based interventions may not have been included in our review due to our search strategy or due to a poor description of the behavioral model in the original paper.

## 5. Conclusions

The results obtained in this review support the use of the TTM in interventions aimed at encouraging PA habits in older adults over 60, considering the TTM and its constructs as a useful tool in the process of creating, developing and evaluating tailored-interventions to promote PA behaviors. However, it is recommended to evaluate the behavioral model in its entirety in order to obtain a more complete and integrated view of behavioral change, in addition, more research is needed to understand the use of PC and their relationship with SoC in the PA behavioral in older adults. Finally, we recommend applying this behavioral model to other groups of older adults with specific characteristics and pathologies in order to determine its validity in that population.

## Figures and Tables

**Figure 1 ijerph-17-09262-f001:**
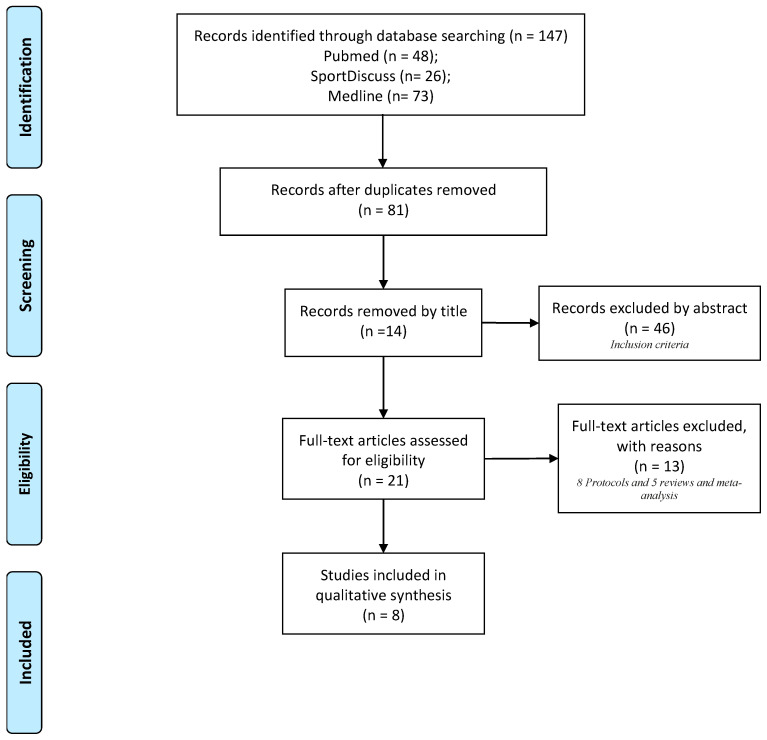
PRISMA flow-chart.

**Table 1 ijerph-17-09262-t001:** Transtheoretical Model (TTM). Description of each construct.

TTM Construct	Description
Stages of change	
Precontemplation	Inactive. No intention to engage in regular PA in the next 6 months.
Contemplation	Inactive but intend to engage in regular PA in the next 6 months.
Preparation	Started to commitment to engage in regular PA.
Action	Established engagement in regular PA but have done so for less than 6 months.
Maintenance	Maintained engagement of regular PA for longer than 6 months.
Process of change	
Cognitive Processes of change	
Consciousness raising (CR)	Increasing knowledge about benefits of PA.
Dramatic relief (DR)	Awareness of health risk of lack PA.
Environmental re-evaluation (ER)	Analyze how the physical inactivity affects to your interpersonal relationships.
Self-reevaluation (SR)	Recognize the benefits of being physically active.
Social liberation (SL)	Show different ways to be active.
Behavioral processes of change	
Counter conditioning (CC)	Change inactive habits for healthy actions.
Helping relationship (HR)	Search social support in sports practice to facilitate change.
Reinforcement management (RM)	Reward yourself for being physically active.
Self-liberation (SeL)	Create commitment to be active and plan activities.
Stimulus control (SC)	Use of stimulus to maintain PA levels.
Decisional Balance	
Pros	Perceived benefits of engaging in regular PA.
Cons	Perceived barriers of engaging in regular PA.
Self-Efficacy	Confidence in one’s ability to perform specific behaviors (regular PA) in specific situations (Tired, in bad mood, do have not time, on vacation and bad weather).

Adapted from Marcus, B.H. and Forsyth, L.H. (2008). Motivating people to be physically active. Human Kinetics [22]. TTM, Transtheoretical model of change; PA, Physical activity; CR, Consciousness raising; DR, Dramatic relief; ER, Environmental re-evaluation; SR, Self-reevaluation; SL, Social liberation; CC, Counter conditioning, HR, Helping relationship; RM, Reinforcement management; SeL, Self-liberation; SC, Stimulus control.

**Table 2 ijerph-17-09262-t002:** TTM constructs.

Reference	Transtheoretical Model Constructs			
	Stages of Change	Processes of Change	Decisional Balance	Self-Efficacy
Bradley, J.M., et al. (2015) [37]	x	x	x	x
Abbaspour, S., et al. (2017) [38]	x		x	x
Koo, K.M., Park, C.H., and Kim, C.J. (2017) [39]	x	x		
Guicciardi, M., et al. (2014) [40]	x			x
Wilson, J.J., et al. (2016) [41]	x	x	x	x
Huang, H. Y., et al. (2015) [42]	x			
Yang, H.J., et al. (2015) [43]	x			
Harada, K., et al. (2014) [44].	x			

TTM, Transtheoretical model of change.

**Table 3 ijerph-17-09262-t003:** TTM in older adults.

Reference	Journal	Country Territory	Study Periods	Study Population	Study Design	Aims	Important Findings
Bradley, J.M., et al. (2015) [37]	BMC Pulmonary Medicine	Ireland		*N*: 55 Age: 63 ± 10 Gender: 22 male (40%) 33 female (60%) Pathology: Bronchiectasis	Descriptive cross-sectional study	Aim: Explore the patterns and correlates of sedentary behavior and PA in bronchiectasis.	- PC: Cognitive and behavioral PC were used equally. - DB: Inverse correlation between the sedentary behavior time and DB ‘pros’ score (*p* = 0.009). - SE: Higher score “when on holiday” (3.35 ± 1.22) and lower score “when I have respiratory symptoms” (1.65 ± 0.97).
Abbaspour, S., Farmanbar, R., Njafi, F., Ghiasvand, A.M., and Dehghankar, L. (2017) [38]	Electronic Physician	Iran	2013	*N*: 262 Age: 64.95 ± 5.03 Gender: 141 male (53.8%) 121 female (46.2%) Pathology: No specific.	Descriptive cross-sectional study	Aim: To identify the relationship between DB and SE in physical activities using the TTM in the members of a retirement center.	- DB: Significant differences between DB and SoC (*p* < 0.001), DB “benefits” and SoC (*p* < 0.0001) and DB “barriers” and SoC (*p* < 0.0001). Significant correlations between “benefits” and PA time (*p* < 0.0001) and significant and reverse association with the “barriers” (*p* < 0.0001). - SE: Significant differences between SE and SoC (*p* < 0.0001). Significative correlations between exercise SE with PA time (*p* < 0.0001). The exercise SE was the only predicator of PA behavior.
Koo, K.M., Park, C.H. and Kim, C.J. (2017) [39]	Journal of Exercise Rehabilitation	Korea	2014	*N*: 81 Age: >60 years Gender: 43 male (53%) 38 female (47%) Pathology: Physical disability or brain injury	Descriptive cross-sectional study	Aim: To develop strategies for promoting PA for the disabled older adults who were in the TTM of precontemplation, contemplation, and preparation stages about participating physical activities for promoting healthy life-styles.	- PC: Significant differences between PC of CR, DR, SR, SL, SeL, and SC with SoC (*p* < 0.05), but no significant differences between PC of ER, CC, HR and RM with the SoC. - PA promotion strategies based on PC were developed for each SoC (precontemplation, contemplation and preparation) for the older adults with disabilities.
Guicciardi, M., Lecis, R., Anziani, C., Corgiolu, L., Porru, A., Pusceddu, M. and Spanu, F. (2014) [40]	Health Psychology and Behavioral Medicine	Italy		*N*: 308 Age: 65.24 ± 8.31 years Gender: 172 male (56%) 136 female (44%) Pathology: Type II Diabetes	Descriptive cross-sectional study	Aim: To investigate the relationships between self-reported PA and exercise SE and body satisfaction in a sample of older adults with Type 2 diabetes classified in different Soc.	- SoC: Significant differences between PA minutes per week and SoC (*p* < 0.001). Significant correlations between SoC and minutes/week of PA (*p* < 0.001) and exercise SE (*p* < 0.001). - SE: Significant differences between exercise SE and SoC (*p* < 0.001). Linear trend for exercise SE across SoC (*p* < 0.001). Significant correlations between exercise SE and minutes/week of PA (*p* < 0.001). - Exercise SE (*p* < 0.001) and SoC (*p* < 0.001) were determinants of PA levels.
Wilson, J.J., Kirk, A., Hayes, K., Bradbury, I., McDonough, S., Tully, M.A., et al. (2016) [41]	Respiratory Care	Ireland		*N*: 55 Age: 63 ± 10 Gender: 22 male (40%) 33 female (60%) Pathology: Non-cystic fibrosis bronchiectasis	Descriptive cross-sectional study	Aim: To examine patterns of (1) PA and (2) mediators of behavior change (SE, DB, and PC) across SoC in individuals with non-cystic fibrosis bronchiectasis.	- SoC: Significant differences between levels of PA with SoC. Significant differences between daily light-lifestyle PA time min/day (*p* = 0.045), daily total PA time (*p* = 0.030), daily total moderate to vigorous PA time (*p*= 0.049) and daily step counts (*p* = 0.03) with SoC. No significant differences in MVPA in 10-min bouts, activity energy expenditure and sedentary behavior time with SoC. - PC: Initial SoC (precontemplation and contemplation stages) used significantly fewer PC compared to advanced SoC (action and maintenance stages). Precontemplation and contemplation stages used significantly more cognitive PC (*p* = 0.031), preparation stage used equally cognitive and behavioral PC (*p* = 0.92) and action and maintenance stages used more behavioral PC (*p* = 0.055). - DB: No significant differences between the DB score (*p* = 0.31) and perceived benefits (*p* = 0.92) with the SoC. - SE: No significant differences between exercise SE and the SoC (*p* = 0.14).
Huang, H.Y., et al. (2015) [42]	Acta Cardiologica Sinica	Taiwan	2010–2011	*N*: 130 Age: 61.0 ± 12.2 years Gender: 92 male (70.8%) 38 female (29.2%) Pathology: Patients after Open Heart Surgery	Design: Prospective cohort study Duration: 6 months.	Aim: To assess exercise behavior and PA levels using TTM in patients undergoing open heart surgery.	- SoC: Significant differences between average of exercise min/week and SoC (*p* = 0.02). 6 months follow-up the inpatient cardiac rehabilitation programs, observed an increase in the percentage of patients in action (39.2%) and maintenance (37.7%) stages.
Yang, H.J., Chen, K.M., Chen, M.D., Wu, H.C., Chang, W.J., Wang, Y.C. and Huang, H.T. (2015) [43]	Journal of Advanced Nursing	Taiwan	2011	*N*: 169 Age: 71.28 ± 5.54 years Gender: 54 male (32%) 115 female (68%) Arms: Control (*n* = 85) Intervention (*n* = 84) Pathology: No specific	Design: Quasi-experimental design Duration: 6 months.	Aim: To test the effects of the group SEB exercises on the functional fitness of community older adults in the contemplation and preparation SoC	- SoC: SoC were used to identify and select participants for an elastic bands exercise program and to evaluate behavior change after 6 months. - Experimental group used strategies own of TTM to facilitate behavioral changes. Experimental group: 86.6% of the participants, switched from contemplation/preparation stages to action stage. Control group: 83.3% of the participants remained in the contemplation/preparation stages after 6 months without training.
Harada, K., Shibata, A., Lee, E., Oka, K. and Nakamura, Y. (2014) [44]	Journal of Physical Activity and Health	Japan	2009	*N*: 1244 Age: 60–74 years Gender: 638 male (51.5%), 600 female (48.5%) Pathology: No specific.	Design: cross-sectional study	Aim: Examined the associations among the perceived health benefits of strength training, perceived barriers to strength straining, and SoC for strength-training behavior.	- SoC: Significant differences between SoC and perceived health benefits (referred to strength exercise recommendations) (*p* <0.0001) and lower perceived barriers (referred to strength exercise recommendations) (*p* < 0.0001). - The perceived benefits and barriers of this article do not correspond to the TTM. Significant differences were observed by gender in perceived benefits (referred to strength exercise recommendations) (men: *p* < 0.001) and (women: *p* < 0.001) and in the perceived barriers (referred to strength exercise recommendations) (men *p* < 0.001) and (women: *p* < 0.001).

TTM, Transtheoretical model of change; PA, Physical activity; SoC, Stages of change; PC, Processes of change, CR, Consciousness raising; DR, Dramatic relief; ER, Environmental re-evaluation; SR, Self-reevaluation; SL, Social liberation; CC, Counter conditioning, HR, Helping relationship; RM, Reinforcement management; SeL, Self-liberation; SC, Stimulus control; DB, Decisional balance; SE, self-efficacy.

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
