# Peer review of "Transtheoretical Model for Physical Activity in Older Adults: Systematic Review"

_ijerph, 2020, doi:10.3390/ijerph17249262_

Round 1

Reviewer 1 Report

Summary: The current manuscript is aimed to systematically review the use of the Transtheoretical model for physical activity in the elderly. Results evidenced that TTM is a useful and suitable behavior model in creating, developing, and evaluating interventions to acquire and improve PA habits in the elderly.

I think the study is well conducted, and I have only a few minor comments.  

Specific comments follow:

Introduction: The introduction fits with the goal of the study. Maybe it is useful to explain some aspects indirectly associated with low physical activity and can be related to problems, including cognitive issues, especially in the elderly population. Such as an increase in BP, the risk of obesity, and an increase in HRV. For a review, see:

Forte, G., & Casagrande, M. (2019). Heart rate variability and cognitive function: a systematic review. Frontiers in neuroscience13, 710.

Favieri, F., & Casagrande, M. (2019). The executive functions in overweight and obesity: a systematic review of neuropsychological cross-sectional and longitudinal studies. Frontiers in Psychology10, 2126.

Forte, G., De Pascalis, V., Favieri, F., & Casagrande, M. (2020). Effects of Blood Pressure on Cognitive Performance: A Systematic Review. Journal of Clinical Medicine9(1), 34.

Methods: Some specific information about the final research script should be added. Moreover, in order to improve the results, a bias risk calculation could be added.

Results: The summary of the study fits the planned analyses. Reorganize the table according to the indications of the journal; moreover, to clarify the reading, the results in the table should be summarized

Discussion: This section could be improved, emphasizing the novelty of this research. It would be interesting to provide more information on the risk factors highlighted (as recommended for the introduction). This, as well as making the results clearer to the reader, could have major clinical implications.

General comment: Generally, I found the study well conducted, although novelty and utility were few emphasized. My advice to the authors is to highlight the innovative nature of the study. This is one of the firsts (presumably the first) study on these fields, and this aspect could be emphasized in discussions. Furthermore, it would be interesting to provide some other implications of results as well as the social implications of this study.

Author Response

We thank the reviewer for his/her comments. We will answer to each comment individually below:

  1. INTRODUCTION: We have added a new reference in the introduction ( Forte, G., & Casagrande, M. (2019). Heart rate variability and cognitive function: a systematic review. Frontiers in neuroscience13, 710)
  2. METHODS: The A final research script is detailed in page 4, at the first paragraph. We did not include it in the manuscript as an independent table/figure.

The research script used in the study was the following:  

1.   Read literature in the topic

From all articles resulting from the search, the abstracts were carefully read, and relevant studies were selected and reviewed in full 

2.     Determinate relevant studies

To determinate a relevant study, at least three of the researches (FJZ, ADS, CR) had to confirm the validity of the theme of the article independently, in case of divergence, a fourth researcher (SAL) was invited to decide whether to include or exclude the studies 

3. Analysis and interpretation of findings for each topic.

4. Evaluation of the quality of the studies (STROBE checklist)

5.  Analysis of the date

6.  Draft to the paper

A bias risk calculation was included in the process because at least three of the researches (FJZ, ADS, CR) had to confirm the validity of the theme of the article independently, in case of divergence, a fourth researcher (SAL) was invited to decide whether to include or exclude the studies. Moreover, the study table (Table 3) was also worked individually by each author and then written together. This aspect has been included in the methodology section.

  1. RESULTS. As suggested the information provided in the tables at the results section, has been reduced. Moreover, the table now follows all indications of the journal.
  2. DISCUSSION. Novelty and utility have been emphasized as suggested. Implications of the results in clinical practice and research have been included. “This is the first study that reviews the use of the TTM for physical activity in pre-frail and frail elderly population. The study encourages the use of TTM for physical activity in clinical practice as part of the elderly health screening at GP practices. Moreover, the use of TTM for physical activity can also lead to more suitable PA programs and/or intervention aimed to improve healthy aging”

Reviewer 2 Report

This article follows a standard systematic review method and is of good quality.

  • On p.3, I recommend clearly naming the components of the model to make the text clearer for the reader. 
  • The first part of the discussion repeats elements that were presented in introduction. I recommend reviewing this section.
  • Language needs to be perfected.

This article would be more relevant if there were clear proposals of elements to include in the guidelines. Presented in the following way, it does not bring much to the field. But it can be published without major changes.

Author Response

We thank the reviewer for his/her time and effort. We have followed the indications and have made the following changes:

  1. The components of the model are clearly named in page 3, with their abbreviation in brackets.
  2. The discussion has been changed including the novelty and utility of the study and the elements repeated from the introduction have been removed.
  3. Language has been revised.

Reviewer 3 Report

The manuscript  elaborated by Fabio Jimenez-Zazo, Susana Aznar and co-workers presents  very interesting systematic review related to transtheoretical model  (TTM) for physical activity (PA) in elderly people.

Although the TTM has  been applied for different health behaviors and in different age groups there is paucity of research in promoting PA in elderly population. The authors used meticulous methods  to identify  all papers related to the TTM based interventions  in adults over 60 year published in English in last 5 years  and to analyze the effect of TTM on the promotion of PA in the elderly population. 

The paper is concise and well written., the references scrupulously selected and analyzed. The authors properly described some limitations of their review. They underline fairly there is paucity of research in this area and only 8 articles from 6 countries met the inclusion criteria with only one prospective cohort study duration of six months.

The authors restricted their systematic review to the papers written in English and published after 2014. I think that a short supplementary information about most important papers related to TTM based PA interventions in elderly population published before 2014 would be justified in discussion in order to increase the substantive value of this interesting manuscript.

Author Response

We thank the reviewer for his/her comments. The systematic review to the papers has been restricted to 2014-onwards because the use of TTM model before was not centered specifically on Physical activity in elderly population.
